# Optimal Algorithm for Max-Min Fair Bandit

Zilong Wang [1]  Zhiyao Zhang [2]  Shuai Li [1]

## Abstract

Multi-player multi-armed bandit (MP-MAB) has been widely studied owing to its diverse applications across numerous domains. We consider an MP-MAB problem where $N$ players compete for $K$ arms in $T$ rounds. The reward distributions are heterogeneous where each player has a different expected reward for the same arm. When multiple players select the same arm, they collide and obtain zero rewards. In this paper, our target is to find the max-min fairness matching that maximizes the reward of the player who receives the lowest reward. This paper improves the existing max-min regret upper bound of $O(\exp(1/\Delta) + K^3 \log T \log \log T)$. More specifically, our decentralized fair elimination algorithm (DFE) deals with heterogeneity and collision carefully and attains a regret upper bound of $O((N^2 + K) \log T/\Delta)$, where $\Delta$ is the minimum reward gap between max-min value and sub-optimal arms. In addition, this paper also provides an $\Omega(\max\{N^2, K\} \log T/\Delta)$ regret lower bound for this problem, which indicates that our algorithm is optimal with respect to key parameters $T, N, K$, and $\Delta$. Additional numerical experiments also show the efficiency and improvement of our algorithms.

## 1. Introduction

Multi-player multi-armed bandit (MP-MAB) problem has been widely studied in recent years due to its wide applications like cognitive radio and wireless network (Wang et al., 2020a; Yang et al., 2022; Wang et al., 2020b; 2022). In such a problem, $N$ players simultaneously play $K$ arms. In each round, each player selects an arm and observes the reward generated from a fixed distribution. In typical MP-MAB problems, players try to maximize the summation of their

[1]Shanghai Jiao Tong University, Shanghai, China [2]The Ohio State University, Columbus, Ohio, USA. Correspondence to: Shuai Li <shuaili8@sjtu.edu.cn>.

*Proceedings of the 42$^{nd}$ International Conference on Machine Learning*, Vancouver, Canada. PMLR 267, 2025. Copyright 2025 by the author(s).

expected reward throughout $T$ rounds. Equivalently, they minimize the total regret, defined as the difference of cumulative reward between their decision and the optimal strategy. MP-MAB problems can be divided into the homogeneous setting and the heterogeneous setting by whether the arm's reward varies among players. Besides, collision is also considered in many applications: when different players select the same arm in one round, they collide and all receive a 0 reward rather than the original rewards drawn from the fixed distribution (Wang et al., 2020b; Boursier & Perchet, 2019; Wang et al., 2020a). For instance, in wireless networks, when more than one users transfer information using one channel, they will interfere with one another and fail, and this is modeled as collisions in our MP-MAB problem.

When considering the heterogeneous setting and collisions, it often raises concerns about unfairness. In order to maximize the total rewards and avoid collisions meanwhile, some players have to sacrifice for the global objective by selecting arms with lower reward (Hossain et al., 2021; Bistritz et al., 2020; Leshem, 2025). Therefore, the primitive objective which only cares about maximizing total rewards, is unfair to those players and makes them have a willingness to not follow the algorithm. Thus researchers hope to search for a better objective to avoid this unfairness. One reasonable objective is to maximize the expected reward of the player who receives the lowest reward, which is called max-min fairness (Bistritz et al., 2020; Leshem, 2025). This ensures that every player is considered equally and nobody should sacrifice. In wireless networks, if only pursuing the maximal Quality of Service (QoS) globally, some users will be annoyed as they can only use channels with low quality to avoid collisions of others.

Bistritz et al. (2020); Leshem (2025) study this max-min fairness topic, but both of them have the following limitations. 1) Their regret upper bounds are larger than $O(\log T)$. The work of Bistritz et al. (2020) derives a regret lower bound of $\Omega(\log T)$. However, existing works only achieve the $O(\log T \log \log T)$ regret upper bound (Bistritz et al., 2020; Leshem, 2025). Additionally, their regret analysis also relies on a large constant, which can grow exponentially with $N$ and $1/\Delta$, where $N$ is the number of players and $\Delta$ is the minimum reward gap. 2) The lower bound $\Omega(\log T/\Delta)$ in Bistritz et al. (2020) only works for parameter $T$. Nevertheless, no lower bound for parameter $N$ or $K$

is provided, which makes the hardness of the problem still unclear. 3) They have some strong assumptions about the setting. Both assume the rewards are bounded, and the latter also assumes $K = N$. Besides, they both assume the reward means of different arms for each player to be different, which is infeasible in many scenarios. These assumptions limit the application scenarios and are of significant interest to be relaxed for wider application.

In order to address the limitations above, we propose our decentralized fair elimination (DFE) algorithm to find the max-min fairness for the decentralized heterogeneous MP-MAB problem where there exists no central server to assign matching directly. We relax the assumptions mentioned before to make our algorithm more general. Besides, it turns out later that this algorithm is optimal since its regret upper bound matches the problem's regret lower bound.

The algorithm makes each player run a phased elimination algorithm. In each phase, players first communicate their current reward estimation to each other. Then all players can compute the lower bound of the max-min value and then eliminate arms whose upper confidence bound (UCB) index is lower than that value. After that, players explore their remaining arms in a round-robin way and the exploration length grows exponentially with the phase. Our decentralized fair elimination algorithm has the following novelties and advantages. 1) Our algorithm adaptively explores sub-optimal arms and eliminates them efficiently, assuring that those sub-optimal arms will not be selected in the later phase. This is the key to improving the regret from $O(\log T \log \log T)$ to the optimal regret of $O(\log T)$. 2) We design a novel exploration assignment algorithm for players at each phase. Due to the elimination procedure, each player may be left with a different set of remaining arms, leading to uneven exploration frequency between different pairs and thus making the exploration process non-trivial. Our DFE algorithm optimizes this process and guarantees that all remaining player-arm pairs will be explored once in at most $N^2 + K - N$ rounds, rather than naive $NK$ rounds.

This paper also shows that one can not find another assignment procedure that achieves better result. 3) A tight $O((N^2 + K) \log T/\Delta)$ regret bound is obtained and this paper also shows that this regret is optimal with respect to all parameters.

More specifically, we also derive a tighter $\Omega(\max\{N^2, K\} \log T/\Delta)$ regret lower bound for this max-min fairness problem. A special case is designed to ensure the regret is lower bounded by $\Delta$ times the selection numbers of specific $\lfloor N/2 \rfloor^2$ sub-optimal player-arm pairs. Then it can be shown that any two of these pairs can not be selected at the same time in case of suffering a large regret. Then we construct another instance that only improves the reward of one of these pairs resulting in a

better max-min value. Thus by divergence decomposition, any pair must be selected $\Omega(\log T/\Delta^2)$, implying the final $\Omega(\max\{N^2, K\} \log T/\Delta)$ lower bound.

**Our Contribution.** Our contributions are outlined as follows.

1. In Section 3, we propose the Decentralized Fair Elimination algorithm (Algorithm 1), which improves the existing result of decentralized case significantly. The regret upper bound of our algorithm is $O((N^2 + K) \log T/\Delta)$. In addition, we relax some assumptions required by previous works in both settings, which allows our algorithm to be applied more widely.

2. In Section 4, we give a tight lower bound, $\Omega(\max\{N^2, K\} \log T/\Delta)$, for max-min MP-MAB problem, which is with respect to not only parameter $T$ proposed in previous work, but also $N$, $K$ and $\Delta$. This lower bound implies that our Algorithm 1 is optimal with respect to all parameters, and thus closes the gap of this problem.

3. In Section 5, numerical experiments are conducted as well. We compare our algorithm with previous works, and these baselines show that our algorithm indeed improves results.

**Related Work.** There is a number of work studying the problem of MP-MAB (Wang et al., 2020a; Yang et al., 2022; Wang et al., 2020b; 2022; Kong & Li, 2023; Kong et al., 2024). They focus on the algorithms' regret, including group regret, individual regret (Wang et al., 2022) and corresponding communication cost (Boursier & Perchet, 2019). Besides, some papers discuss the homogeneous and the more general heterogeneous settings of MP-MAB and propose algorithms for them respectively(Yang et al., 2022; Magesh & Veeravalli, 2022). On the other hand, the existence or absence of central server are discussed in Shi et al. (2021); Mehrabian et al. (2020); Buccapatnam et al. (2015); Kolla et al. (2018). However, the fairness problem is ignored in these scenarios. As we discussed above, the potential risk of unfairness for sacrificed players is of great significance to be studied, and this paper proposes an efficient algorithm to fill this gap.

Regarding the fairness problem, there are many kinds of fairness. Some are from the aspect of arms (Joseph et al., 2016; Wang et al., 2021; Fang et al., 2022; Mansoury et al., 2024; Jeunen & Goethals, 2021), and some are for players (Hossain et al., 2021; Bistritz et al., 2020; Leshem, 2025). Max-min fairness is a kind of fairness for players and is widely considered in wireless networks. Although Bistritz et al. (2020); Leshem (2025) have proposed some algorithms to handle unfairness in max-min MP-MAB, their assumptions and results are still not tight and need to be improved,

as we have discussed before. Our near-optimal algorithms with tight bounds for all parameters dramatically improve the results.

## 2. Preliminaries

This paper considers the multi-player multi-armed bandit problem consisting of $N$ players and $K$ arms, denoted as sets $\mathcal{N} := \{1, \ldots, N\}$ and $\mathcal{K} := \{1, \ldots, K\}$, respectively. We assume $N \leq K$ to ensure that all players are able to select their arms if no collision happens. There are $T$ rounds and each player $i \in \mathcal{N}$ selects an arm $k_i(t) \in \mathcal{K}$ and receives a reward $r_{i,k}(t)$ in each round $t \in [T]$. Denote all players' selections in round $t$ by the matching set $m(t) := \{(1, k_1(t)), \ldots, (N, k_N(t))\}$. When multiple players select the same arm simultaneously, a collision occurs between these players and the reward of these colliding players is $r_{i,k_i(t)}(t) = 0$; otherwise, the reward $r_{i,k_i(t)}(t)$ is an i.i.d. random variable generated from a 1-sub-gaussian distribution with mean $\mu_{i,k_i(t)} \geq 0$. The reward means are heterogeneous, i.e., $\mu_{i,k}$ can be different from $\mu_{i',k}$ for $i \neq i'$. Denote the collision indicator for player $i$ in round $t$ by $C_i(t) = \mathbb{1}\{\text{player } i \text{ suffers a collision in round } t\}$. This paper considers the decentralized setting where there is no central platform to assign matching. Each player $i$ selects the arm based on its own history observations $\{r_{s,k_i(s)}(s), C_i(s)\}_{s \in [t-1]}$.

In the multi-player bandit literature, max-min fairness is of great importance as it maximizes the minimum resource share, ensuring fairness and system stability among multiple entities. The fairness objective of players is to find the max-min value, i.e., to maximize the minimum reward in a matching $m$, denoted by $\gamma(m) = \min_i \mu_{i,m_i}$, where $m_i$ is the arm selected by player $i$ in matching $m$. We define the optimal max-min value $\gamma^*$ and optimal max-min matching $m^*$ by

$$\gamma^* = \max_m \min_i \mu_{i,m_i}, \quad m^* \in \arg\max_m \min_i \mu_{i,m_i}.$$

The regret is defined as the difference between the optimal max-min reward $\gamma^*$ and the minimum expected reward among the selected arms at each round $t$, same as (Bistritz et al., 2020; Leshem, 2025):

$$R(T) = \mathbb{E}\left[\sum_{t=1}^{T}\left(\gamma^* - \min_i\left\{(1 - C_i(t)) \cdot \mu_{i,k_i(t)}\right\}\right)\right].$$

## 3. Decentralized Fair Elimination Algorithm

In this section, we introduce the Decentralized Fair Elimination algorithm (Algorithm 1).

In general, the Decentralized Fair Elimination algorithm (Algorithm 1) executes by epochs. In each epoch $s$, Algorithm 1 has three phases: elimination (Lines 3 - 12),

---

**Algorithm 1** Decentralized Fair Elimination (for player $i$)

1: Initialize: $\hat{\mu}_{i,k}(0) = 0, N_{i,k}(0) = 0, \text{UCB}_{i,k}(0) = \infty, \text{LCB}_{i,k}(0) = -\infty, \forall i \in [N], k \in [K]; \mathcal{P} = \{(i,k) \mid \forall i \in [N], k \in [K]\}$.
2: **for** phase $s = 1, 2, \cdots$ **do**
3:     Set $\mathcal{M}_s = \emptyset$.
4:     Compute max-min matching based on $\{\text{LCB}_{i,k}(s-1)\}_{i \in [N], k \in [K]}$, obtain max-min value $\underline{\gamma}_s$;
5:     **for** $\forall j \in [N], k \in [K]$ **do**
6:       **if** $\text{UCB}_{j,k}(s-1) < \underline{\gamma}_s$ **then**
7:         $\mathcal{P} = \mathcal{P} \backslash \{(j,k)\}$;
8:       **end if**
9:       **if** there does not exist matching $m$ which contains $(j,k)$ s.t. $\min_i \text{UCB}_{i,m_i}(s) > \underline{\gamma}_s$ **then**
10:         $\mathcal{P} = \mathcal{P} \backslash \{(j,k)\}$;
11:       **end if**
12:     **end for**
13:     $\mathcal{M}_s = $ **Assign Exploration**$(\mathcal{P})$ by Algorithm 2;
14:     **for** $m \in \mathcal{M}_s$ **do**
15:       Select $m_i$ for $2^s$ times;
16:       Update $\hat{\mu}_{i,k}(s), N_{i,k}(s), \text{UCB}_{i,k}(s), \text{LCB}_{i,k}(s), \forall i \in [N], k \in [K]$;
17:     **end for**
18:     Communicate by Algorithm 3, get information $\text{UCB}_{i',k}(s), \text{LCB}_{i',k}(s)$ for any player $i'$;
19: **end for**

---

exploration (Lines 13 - 17), and communication (Line 18). In the elimination phase, players compute the lower bound of max-min matching based on the observed information and eliminate those sub-optimal arms; in the exploration phase, players explore their corresponding non-eliminating arms in a round-robin way. At last, players share their local information during the communication phase.

### 3.1. Elimination Phase

In the $s$-th elimination phase, all players follow the same rule to compute the lower bound of current max-min matching and to eliminate sub-optimal arms based on the latest information derived from the communication phase. Each player $i$ computes $\text{LCB}_{j,k}(s) = \hat{\mu}_{i,k}(s) - \sqrt{\frac{6 \log T}{N_{i,k}(s)}}$ and $\text{UCB}_{j,k}(s) = \hat{\mu}_{i,k}(s) + \sqrt{\frac{6 \log T}{N_{i,k}(s)}}$ for all $j \in \mathcal{N}, k \in \mathcal{K}$. Then, players compute the max-min matching and corresponding max-min value $\underline{\gamma}_s$ with respect to LCBs (Line 4).

The max-min matching can be found through a threshold-based algorithm (Panagiotas et al., 2023). First, sort all LCB indexes $\{\text{LCB}_{i,k}(s)\}_{i \in [N], k \in [K]}$ in decreasing order, and set as threshold $\gamma'$ in order of sorting (or we can apply binary search to set threshold for reducing time complexity),

testing if one can construct a perfect matching only using player-arm LCB index higher than the threshold, i.e., construct a perfect matching $m$ with $\text{LCB}_{i,m_i}(s) \geq \gamma'$. The maximum threshold that can form a perfect matching is the corresponding max-min value with respect to LCBs, denoted as $\underline{\gamma}_s$. For finding the max-min matching we can apply the Hungarian algorithm (Algorithm 5 in Leshem (2025)). The total time complexity is $O(\text{poly}(N, K))$, and we remark that this can be solved offline and does not affect the online learning efficiency.

After that, players eliminate the player-arm pairs who can not form a matching with minimum UCB value larger than the computed lower bound of max-min value (Lines 5 - 12). Denote $\mathcal{P}$ as the remaining non-eliminating player-arm set. If $\text{UCB}_{j,k}(s) < \underline{\gamma}_s$, then pair $(j, k)$ will be eliminated from the set $\mathcal{P}$ (Lines 6 - 8). Additionally, if for pair $(j, k)$, we can not construct a perfect matching $m$ containing $(j, k)$ with minimum UCB index $\min_i \text{UCB}_{i,m_i}(s)$ greater than $\underline{\gamma}_s$, $(j, k)$ will also be eliminated from $\mathcal{P}$ since with high probability $(j, k)$ will not occur in the optimal max-min matching $m^*$ (Lines 9 - 11). The details of elimination phase are described in Algorithm 1.

### 3.2. Exploration Phase

The elimination phase is followed by the exploration phase in each epoch. Here we introduce our new assignment algorithm (Algorithm 2), which only takes at most $N^2 + K$ rounds to explore all non-eliminating pairs once under the constraint of not selecting eliminated pairs, shown to be the optimal assignment method in Section 4.

In the $s$-th exploration phase, the exploration of all player-arm pairs within $\mathcal{P}$ remains necessary. By the eliminating rule, it is established that for each pair $(i, k) \in \mathcal{P}$, there must exist a corresponding matching $m$ such that every pair within $m$ also belongs to $\mathcal{P}$, which can be precisely denoted as $\forall (i, k) \in \mathcal{P}, \exists m, m_i = k, \forall (i', k') \in m, (i', k') \in \mathcal{P}$.

Consequently, a matching $m'$ can be initially located such that all pairs within $m'$ are elements of $\mathcal{P}$. Subsequently, an arm set $\mathcal{K}_{m'} = \{m'_1, \ldots, m'_N\}$, which comprises all the arms within $m'$, is obtained (Lines 2 - 3). For any pair $(i, k)$ in $\mathcal{P}$, if $k \in \mathcal{K}_{m'}$, a matching $m$ is constructed with $m_i = k$ and $\forall i' \in [N], (i', m_{i'}) \in \mathcal{P}$. Each such $m$ is then appended to $\mathcal{M}$, and during this construction, at most $N^2$ matchings are generated (Lines 4 - 9).

Afterward, for the remaining $K - N$ arms that are not included in $\mathcal{K}_{m'}$, new indexes ranging from $N + 1$ to $K$ are assigned (Line 10). That is, the index of $m'_i$ is extended to $m'_K$ in such a manner that each arm $k \in [K]$ has a unique index $u \in [K]$ satisfying $m'_u = k$. To explore those pairs within $\mathcal{P}$ whose arms are not in $\mathcal{K}_{m'}$, $K - N$ matchings can be constructed to explore those arms in a round-robin

way. Specifically, in the $r$-th matching, player $i$ will select $m'_{N+((i+r) \bmod (K-N))}$ if the corresponding pair is in $\mathcal{P}$; otherwise, player $i$ will select $m'_i$ (Lines 11 - 12). Through this design, it can be ensured that all pairs within $\mathcal{P}$ will be explored precisely once within at most $N^2 + K$ rounds, and moreover, all the eliminated pairs will not be selected. In the $s$-th exploration phase, all players will select each matching for $2^s$ rounds.

Figure 1 provides an illustrative example to explain the exploration phase, in which blue dashed circles represent eliminated player pairs, i.e., $(i, k) \notin \mathcal{P}$. Other solid circles are available, i.e., $(i, k) \in \mathcal{P}$. When exploring pairs to the left of the vertical dashed line (arms in $\mathcal{K}_{m'}$), players can cover all of them once in at most $N^2$ rounds. When exploring pairs to the right of the vertical dashed line in a round-robin way, players select the corresponding red pair in its line if they meet a blank, i.e., an eliminated pair. For example, in Figure 1 player 2 has eliminated arm $m_{N+1}$, so when it is her turn to select $m_{N+1}$ in round-robin exploration, it will select $m_2$ instead to avoid selecting eliminated arms. Therefore, these pairs can be explored once with no more than $K - N$ rounds.

Compared with naively constructing a matching for every pair in $\mathcal{P}$ which needs at most $NK$ matchings, we improve this upper bound to $N^2 + K$. Later in the instance shown in the lower bound analysis, it can be seen that there exists a pair set $\mathcal{P}$ such that the number of times for selecting all pairs once is lower bounded by $\Omega(N^2 + K)$ rounds if not selecting pairs out of $\mathcal{P}$. This shows our assignment algorithm is near optimal.

Note that if we relax the constraint that lets players select eliminated arms, i.e, pairs out of $\mathcal{P}$, a round-robin exploration strategy can be directly used to explore all arms in $K$ rounds. However, we make such a constraint that avoids selecting any eliminated pair in the exploration phase since selecting the eliminated pair leads to more regret in one round. For those eliminated arms, they have been identified as sub-optimal and usually with low reward, which means there will be large regret if selecting the eliminated arm in the exploration phase. In the regret analysis we can see that by avoiding exploring the eliminated arm, the regret can be improved from $O(\log T / \Delta^2)$ to $O(\log T / \Delta)$, where $\Delta$ is the minimum reward gap between $\gamma^*$. This shows our algorithm outperforms in the scenario where there are pairs with rewards near $\gamma^*$.

### 3.3. Communication Phase

Note that in this paper we study the problem in decentralized setting where each player can only observe its own reward and make decisions based on its historical observation. This setting is extensively studied by MP-MAB works. Furthermore, communication process among players is unavoidable

---

**Algorithm 2** Assign Exploration

---

**Input:** Non-eliminating player-arm set $\mathcal{P}$.
 1: Initialize $\mathcal{M} = \emptyset$;
 2: Find a matching $m'$ satisfying $\forall (i,k) \in m', (i,k) \in \mathcal{P}$;
 3: $\mathcal{M} = \mathcal{M} \cup \{m'\}$. Denote $\mathcal{K}_{m'} = \{m'_i \mid i \in [N]\}$;
 4: **for** $\forall (i,k) \in \mathcal{P}$ **do**
 5:  **if** $k \in \mathcal{P}$ **then**
 6:   Construct a matching $m$ with $m_i = k$ and $\forall (i',k) \in m, (i',k) \in \mathcal{P}$;
 7:   $\mathcal{M} = \mathcal{M} \cup \{m\}$;
 8:  **end if**
 9: **end for**
10: Extend the index of $m'_i$ to $m'_K$ such that each arm $k \in [K]$ has its unique corresponding index $u$ satisfying $m'_u = k$;
11: Construct $K - N$ matching such that in the $r$-th matching $m^r$, player $i$ selects

$$m_i^r = \begin{cases} m'_{N+((i+r) \mod (K-N))}, & (i, m'_{N+((i+r) \mod (K-N))}) \in \mathcal{P} \\ m'_i, & (i, m'_{N+((i+r) \mod (K-N))}) \notin \mathcal{P} \end{cases}$$

12: $\mathcal{M} = \mathcal{M} \cup \{m^r\}$ for $\forall r \in [K-N]$;
**Output:** Exploration matching set $\mathcal{M}$;

---

since all players have to agree on a matching to avoid collisions. In this paper we control the exponential growth of communication intervals, guaranteeing that the total communication times is bounded by $O(\log T)$. Here, we assume that at each round a player can transmit estimated reward information of one arm to another player with regret $\gamma^*$, where $\gamma^*$ is the optimal max-min value. This assumption is mild since the amount of information transmitted in one round is bounded. Moreover, this assumption can also be removed by making players transfer the information into binary and communicate by colliding on bit "1" (Boursier & Perchet, 2019) (Remark 1).

The communication phase has a fixed length. In $s$-phase communication, player $i$ sends $\hat{\mu}_{i,k}(s)$ for $k \in [K]$ to all other $N - 1$ players, and receives $\hat{\mu}_{j,k}(s)$ from each player $j \in [N]$ for $k \in [K]$. As $K \geq N$, there are at most $\lfloor \frac{N}{2} \rfloor$ pairs of players exchanging information meanwhile. The order of the exchange can be fixed before the game, which only needs to know the number and index of players. Here we assume that any two players can exchange their estimations of reward in one round with regret $\gamma^*$ (maximal regret in one round). Therefore, each communication phase has a fixed length of $N$.

### 3.4. Analysis and Discussion

Now, we are ready to state our main result as the next theorem. The proof is deferred in Appendix B.

**Theorem 3.1.** *For max-min MPMAB problem with $N$ players, $K$ arms, and time horizon $T$. Let each player play according to Algorithm 1, the total expected regret is bounded*

*by*

$$\mathbb{E}[R(T)] \leq 164(N^2 + K)\log T/\Delta + \gamma^* N \log T + 2NK,$$

*where $\Delta := \min_{i,k:\mu_{i,k} < \gamma^*} \{\gamma^* - \mu_{i,k}\}$ and $\gamma^*$ is the max-min value.*

*Remark 1.* We assume that each player can communicate with another player in one round with regret $\gamma^*$. This assumption simplifies the analysis of the communication phase. Without this assumption, we could still use collisions to transmit information bit by bit, resulting in an additional constant length of information bits. The details of this process can be found in Appendix F. If the minimum reward gap between max-min value $\gamma^*$ is $\Delta$, then the length of each communication phase is bounded by $N \log(1/\Delta)$. Here, $\log(1/\Delta)$ is the length of transmitting a reward's information by bit and through collisions. We only need the bit length with $\log(1/\Delta)$ since it is enough to distinguish two pairs with gap larger than $\Delta$. Then the total communication regret is bounded by $N \log(1/\Delta) \log T$. We also note that the minimum communication cost studied in previous work is $\frac{3}{2} N^3 \log(1/\Delta) \log T$ (Leshem, 2025). Additionally, we note that $\Delta$ in their works is the minimum reward gap among all player-arm pairs, whereas in our work $\Delta$ is only the minimum reward gap between $\gamma^*$. Thus we also significantly improve the communication cost compared with previous works.

*Remark 2.* We highlight that this exploration design is optimal with order $O(N^2 + K)$. In Section 4 we will see that the given special case provides a $\Omega(N^2 + K)$ assignment lower bound when the player-arm pairs with 0 reward are all eliminated. Thus our method indeed gives an optimal so-

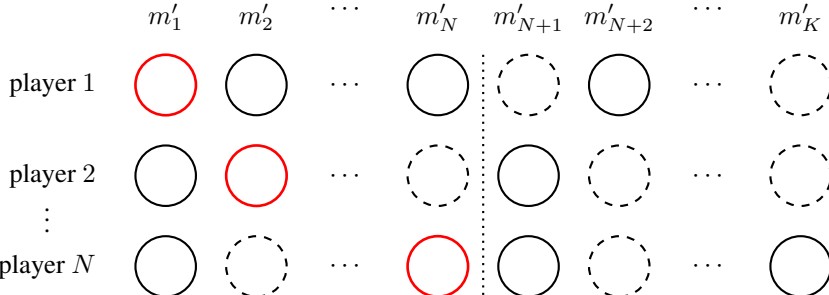

**Figure 1:** Example for exploration phase. For the left $N$ arms, we construct the matching for each $(i, k)$ pairs, requiring at most $N^2$ matchings. For the right $K - N$ arms, we apply the round-robin methods and replace those eliminated arms with corresponding red solid arms to avoid collisions.

lution for such a player-arm pair covering problem without collisions. Besides, this technique can also be considered to apply for other MP-MAB problems with heterogeneous reward, which requires different exploration times for each player-arm pair to reach the optimal regret.

*Remark 3.* Note that previous works (Bistritz et al., 2020; Leshem, 2025) also provide a phased-based algorithm to deal with the decentralized setting. However, they both apply an explore-then-commit (ETC) method at each epoch $s$. Specifically, they let each player explore each arm $\log s$ times at the beginning of the epoch $s$, and then compute the max-min matching based on history observations in exploration phase. After that each player follows this matching in the following $2^s$ rounds. Their algorithms both only obtain an $O(\log T \log \log T)$ regret bound since they have to set an increasing length of exploration at each epoch. This design is to make sure the probability of computing a wrong max-min matching is bounded by $\exp(-s)$ when $s$ is sufficiently large that $\log s > 1/\Delta$, then the regret in the exploitation phase can be bounded. This design also raises the problem of a large constant to guarantee $\log s > 1/\Delta$, which requires initial warm-up rounds is $O(\exp(1/\Delta))$, which could be very large when $\Delta$ is small enough. We handle this problem by applying the elimination method which eliminate sub-optimal player-arm pair efficiently. This assures that no forced explorations will happen in later epochs.

*Remark 4.* We claim that in the beginning of the game, there will be an index searching process if players do not know their indices beforehand. We note that this is a standard process in decentralized MP-MAP problem (Rosenski et al., 2016; Boursier & Perchet, 2019) and we can run the process proposed in (Leshem, 2025). In this algorithm, players determine their respective indices according to a randomly choosing arm method. They show that this process will end in less than $O(N \log N)$ rounds with probability 1, therefore, the regret of this process is also less than $O(N \log N)$. Since the regret of by index searching process never dominates, for the convenience of description, we ignore it when

considering the overall regret.

## 4. Regret Lower Bound

In this section, we present a regret lower bound analysis for max-min MP-MAB problems, which shows that the regret of the DFE algorithm (Algorithm 1) is tight with respect to parameters $K, N, \Delta$ and $T$.

Let $R(T, \nu, \pi)$ denote the expected regret of a policy $\pi$ on the instance with an arm distributions $\nu = \{\nu_{i,k} : i \in [N], k \in [K]\}$ for a horizon of length $T$. Denote $\mathcal{P}$ as the set of all probability distributions of reward bounded by $[0, 1]$.

Define a policy is *uniformly consistent* if and only if for all $\nu \in \mathcal{P}$, all $\alpha \in (0, 1)$, the regret $\limsup_{T \to \infty} \frac{R(T, \nu, \pi)}{T^\alpha} = 0$. This notion is used to eliminate tuning a policy to the current instance while admitting large regret in other instances. In this paper, a tight lower bound result is derived in the following theorem.

**Theorem 4.1.** *For max-min MPMAB problem with $N$ players, $K$ arms, and time horizon $T$, there exists a instance $\nu$ with minimum gap $\Delta < 1/N$ and the reward distributions are Bernoulli, for any uniformly consistent policy $\pi$ satisfies*

$$R(T, \nu, \pi) \geq \sum_{i=N-\lfloor \frac{N}{2} \rfloor+1}^{N} \sum_{k=1}^{\lfloor \frac{N}{2} \rfloor} \frac{\log T}{\Delta}.$$

Theorem 4.1 states that for max-min MP-MAB problem with $N$ players, $K$ arms, and time horizon $T$, there exists an instance that for any uniformly consistent policy, for minimum gap $\Delta < 1/N$, it takes the regret at least

$$\Omega(\max\{N^2, K\} \log T/\Delta),$$

where $\Delta = \min_{(i,k):\gamma^* > \mu_{i,k}}\{\gamma^* - \mu_{i,k}\}$.

Then, we provide the proof sketch of Theorem 4.1. The detail of the lower bound proof is deferred to Appendix C.

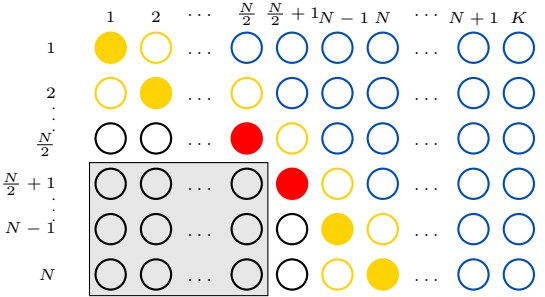

**Figure 2:** Base case when $N$ is even and solid circles represent max-min matching. The yellow cycle represents the reward of $\frac{1}{2} + 2\Delta$, the red cycle represents the reward of $\frac{1}{2} + \Delta$, the black cycle represents the reward of $\frac{1}{2}$, and the blue cycle represents the reward of $0$.

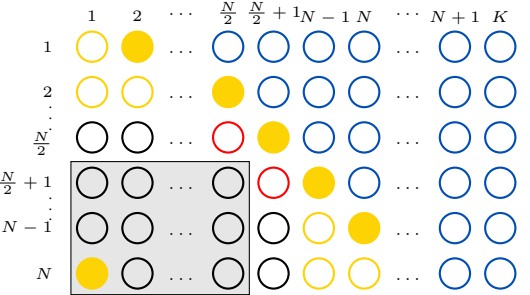

**Figure 3:** Increasing the value of the pair $(N, 1)$ to be optimal, the max-min matching changes. The solid circles represent the corresponding max-min matching.

We first construct the special instance $\nu$ with $N < K$, the reward mean is designed as follows:

- For player $i \in [1, \lfloor \frac{N}{2} \rfloor - 1]$, the mean reward of arm $k \in [1, i+1]$ is $1/2 + 2\Delta$, the mean reward of $k \in [i+2, K]$ is $0$.

- For player $i \in [\lfloor \frac{N}{2} \rfloor, N - \lfloor \frac{N}{2} \rfloor + 1]$, the mean reward of arm $k \in [1, i-1]$ is $1/2$, the mean reward of arm $k = i$ is $1/2 + \Delta$, the mean reward of arm $k = i+1$ is $1/2 + 2\Delta$, and the mean reward of arm $k \in [i+2, K]$ is $0$.

- For player $i \in [N - \lfloor \frac{N}{2} \rfloor + 2, N]$, the mean reward of arm $k \in [1, N - \lfloor \frac{N}{2} \rfloor + 2]$ is $1/2$, the mean reward of arm $k \in [N - \lfloor \frac{N}{2} \rfloor + 3, \min\{i+1, N\}]$ is $\frac{1}{2} + 2\Delta$, and the mean reward of $k \in [\min\{i+1, N\} + 1, K]$ is $0$.

Denote $\nu_{i,k}$ as the distribution of rewards obtained when arm $k$ is matched to player $i$ in this environment. The special case is shown in Figures 2 and 3.

For this instance, it can be shown that the max-min value $\gamma^* = \frac{1}{2} + \Delta$ with corresponding max-min matching

$m^* = \{(1,1), (2,2), \ldots, (N,N)\}$, shown in Figure 2: First, matching $m^*$ has the minimum reward $\frac{1}{2} + \Delta$. Second, if there exists a matching $m'$ with minimum reward $\frac{1}{2} + 2\Delta$, then by construction the player $i \in [\lfloor \frac{N}{2} \rfloor + 1, N - \lfloor \frac{N}{2} \rfloor + 1]$ must select arm $k = i + 1$. To reach the reward $\frac{1}{2} + 2\Delta$ player $i \in [N - \lfloor \frac{N}{2} \rfloor + 2, N - 1]$ must select arm $i + 1$, and then player $i = N$ fails to select any arm with reward $\frac{1}{2} + 2\Delta$. Thus the max-min value is $\frac{1}{2} + \Delta$.

Denote the set of player-arm pairs $S_1 = \{(i,k) \mid \mu_{i,k} = 0\}$, $S_2 = \{(i,k) \mid i \in [N - \lfloor \frac{N}{2} \rfloor + 1, N], k \in [1, \lfloor \frac{N}{2} \rfloor]]\}$ (gray shaded area in Figure 2). Selecting a player-arm pair in $S_1$ will suffer a large constant regret $\frac{1}{2} + \Delta$. Note that when not selecting any pair in $S_1$, any two pairs in $S_2$ will not be in the same matching, as stated in the following claim:

*Claim* 1. Given a matching $m$, if $\forall (i,k) \in S_1$, $(i,k) \notin m$, then $\forall (i,k), (i',k') \in S_2$ with $(i,k) \neq (i',k')$, we have $(i,k) \notin m$ or $(i',k') \notin m$.

*Proof.* We show this claim by contradiction. Consider a matching $m$ that $\forall (i,k) \in S_1$, $(i,k) \notin m$. Suppose $\exists (i,k), (i',k') \in m$ with $i < i'$, $(i,k) \in S_2$ and $(i',k') \in S_2$. From the construction of $S_1$ and instance $\nu$, we have that for player $i'' \leq \lfloor \frac{N}{2} \rfloor$, it can only select arm with index less than $\lfloor \frac{N}{2} \rfloor + 1$. Since $i' > i > \lfloor \frac{N}{2} \rfloor$, the number of remaining arms available for top $\lfloor \frac{N}{2} \rfloor$ players is $\lfloor \frac{N}{2} \rfloor - 1$. This contradicts that $m$ is a matching. $\square$

This claim shows that in order to explore all pairs in $S_2$ without selecting low reward pairs in $S_1$, it is necessary to construct matching for each pair in $S_2$ separately. In other words, at least $\lfloor \frac{N}{2} \rfloor^2$ matchings are needed in exploration.

Then we can derive the following lemma, which is proved in Appendix D:

**Lemma 4.2.** *For the above instance $\nu$, and any uniformly consistent policy $\pi$, for $\Delta < 1/N$, the following inequality holds:*

$$R(T, \nu, \pi) \geq \Delta \sum_{i=N-\lfloor \frac{N}{2} \rfloor + 1}^{N} \sum_{k=1}^{\lfloor \frac{N}{2} \rfloor} \mathbb{E}\left[N_{i,k}(T)\right],$$

*where $N_{i,k}(T)$ is the number of times $(i,k)$ is selected up to time $t$.*

For any pair $(i,k)$ in $S_2$, we can show that when $\mu_{i,k}$ is changed to $\mu'_{i,k}$ higher than $\frac{1}{2} + \Delta$, then the max-min value will be changed to $\mu'_{i,k}$. Figure 3 shows the corresponding max-min matching in this instance. This motivates us to design another instance $\nu'$ which only changes the distribution of $(i,k)$. Then applying the divergence decomposition technique to lower bound $N_{i,k}(T)$ Theorem 4.1 is obtained.

*Remark.* Note that Bistritz et al. (2020) first derived an $\Omega(\log T/\Delta)$ lower bound of max-min problem. However,

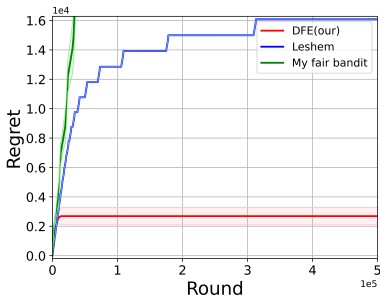
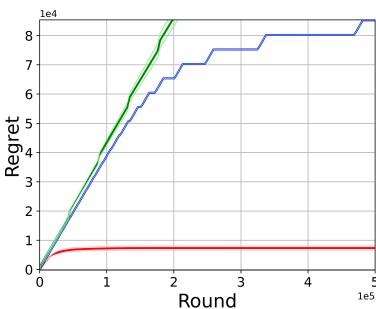
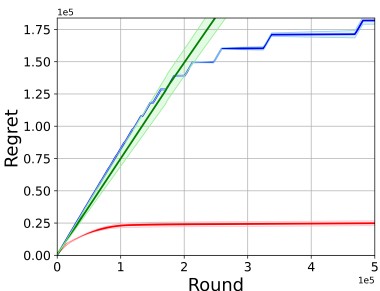

**(a)** Cumulative regrets for the DFE algorithm, My Fair Bandit algorithm and the Leshem algorithm with $N = 4, K = 4$.

**(b)** Cumulative regrets for the DFE algorithm, My Fair Bandit algorithm and the Leshem algorithm with $N = 10, K = 10$.

**(c)** Cumulative regrets for the DFE algorithm, My Fair Bandit algorithm and the Leshem algorithm with $N = 10, K = 15$.

**Figure 4:** Experimental comparisons of our DFE algorithm with Leshem (Leshem, 2025) and My Fair Bandit algorithms (Bistritz et al., 2020) under $(N, K) = (4, 4), (10, 10), (10, 15)$ settings respectively.

that is a loose result that does not underscore the roles of $K$ and $N$. By contrast, we give a tighter lower bound in Theorem 4.1, which works for all related parameters. In Section 3, the regret of the algorithm proposed in this paper matches such lower bound on all parameters. It means that this paper not only improves the theoretical result, but also closes the gap of this problem.

## 5. Experiments

This section provides numerical simulations to validate that our DFE algorithm performs well in max-min MP-MAB problems. Besides, the comparison with previous works verifies significant improvements of our algorithm.

Here, we take $T = 500,000$, and change the values of $N$ and $K$ to conduct multiple experiments: $(N, K) = (4, 4), (10, 10), (10, 15)$. The reward of each player-arm pair $(i, k)$ follows a Gaussian distribution $\mathcal{N}(\mu_{i,k}, \sigma^2)$ with $\sigma = 1$. The reward means form a reward matrix $U$, whose element in the $i$-th row and the $k$-th column is $\mu_{i,k}$. The reward matrix of $(4, 4)$ and $(10, 10)$ are the same with which in Bistritz et al. (2020), shown in Appendix E. And for $(10, 15)$, the mean value is generated uniformly from $[0, 1]$. We conduct experiments with three algorithms for comparison: DFE (Algorithm 1), Leshem ((Leshem, 2025)), and My Fair Bandit ((Bistritz et al., 2020)). Each experiment is repeated 20 times. All plots are averaged over 20 trials with confidence intervals of 95%.

Figure 4 shows that our DFE algorithm dramatically decreases the regret compared to the other two algorithms. Specifically, in all three experiments, The regret of DFE algorithm decreases by more than 90% of My Fair Bandit algorithm and more than 80% of the Leshem algorithm. In addition, the DFE algorithm converges quickly. On the other hand, the other two algorithms may experience a long time

before converging, as Figure 4b shows. Besides, the other two algorithms both suffer regret continuously since they apply the phased ETC method which requires lasting explorations even though each player has enough exploration, while our algorithm would stop selecting sub-optimal pairs when they are eliminated. At last, note that our novel exploration assignment method which explores not eliminated pairs nearly uniformly also helps the algorithm attain less regret, as the other two algorithms simply explore all arms even if they have been identified to be sub-optimal. These numerical results verify the advantage of our algorithm.

## 6. Conclusion

This paper discusses the max-min MP-MAB problem and proposes a new algorithm with a tight regret upper bound, which effectively solves the problem of unfairness. Specifically, the decentralized fair elimination (DFE) algorithm is proposed and attains the $O((N^2 + K) \log T/\Delta)$ regret upper bound to achieve the max-min fairness. This result significantly improves the previous regret bound of $O(\exp(1/\Delta) + K^3 \log T \log \log T)$ with respect to all parameters (Leshem, 2025). Moreover, a tighter regret lower bound of $\Omega(\max\{N^2, K\} \log T/\Delta)$ is derived, which shows our proposed algorithm exactly matches the lower bound. Additional experiments also show the efficiency and improvements of our DFE algorithm.

One future direction of our work is to focus on the robustness and incentive compatibility of the max-min MP-MAB problem. Another promising future direction is that our insights into the exploration stage in our DFE algorithm can be generalized into other kinds of MP-MAB problems, which serves as an effective technique dealing with collisions.

## Acknowledgements

The corresponding author Shuai Li is supported by National Natural Science Foundation of China (62376154).

## Impact Statement

This paper presents work whose goal is to advance the field of Online Learning. There are many potential societal consequences of our work, none which we feel must be specifically highlighted here.

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

# Appendix

## A. Notation

In this section, we summarize all used notations with their meanings in the following table.

| | |
|---|---|
| $N$ | Number of players |
| $K$ | Number of arms |
| $T$ | Number of rounds |
| $k_i(t)$ | Arm player $i$ selects at round $t$ |
| $r_{i,k}(t)$ | Reward of player $i$ selecting arm $k$ at round $t$ |
| $\mu_{i,k}$ | Mean of reward player $i$ selecting arm $j$ |
| $C_i(t)$ | Collision indicator of player $i$ at time $t$ |
| $m$ | Matching |
| $\mathcal{M}$ | Set of matching |
| $\gamma(m)$ | Minimum mean reward in matching $m$ |
| $\gamma^*$ | Max-min mean reward |
| $m^*$ | Matching with max-min mean reward |
| $\hat{\mu}_{i,k}(t)$ | Empirical mean reward of $i$ selecting $k$ at $t$ |
| $N_{i,k}(t)$ | Number of times player $i$ selects arm $k$ up $t$ |
| $\text{UCB}_{i,k}(t)$ | UCB index of player $i$ for arm $k$ at round $t$ |
| $\text{LCB}_{i,k}(t)$ | LCB index of player $i$ for arm $k$ at round $t$ |
| $\underline{\gamma}_s(m)$ | Minimum LCB value at phase $s$ in matching $m$ |
| $\overline{\gamma}_s(m)$ | Minimum UCB value at phase $s$ in matching $m$ |
| $\underline{\gamma}_s$ | Max-min LCB index at phase $s$ |

**Table 1:** Notation table that includes all used notations with their meanings in the paper.

## B. Proof for Section 3

In this section, we first state the supplementary lemma (proved in Appendix D) for Theorem 3.1, and then prove this theorem.

**Lemma B.1.** *Let $\mathcal{F} = \{\exists i \in [N], \exists k \in [K], |\hat{\mu}_{i,k}(t) - \mu_{i,k}| > \sqrt{\frac{6\log(T)}{N_{i,k}(t)}}\}$ be the bad event that some player-arm rewards are not estimated well at time $t$. We have:*

$$\mathbb{P}(\mathcal{F}) \leq 2NK/T .$$

### B.1. Proof for Theorem 3.1

*Proof.* First, from algorithm design we have that the regret is caused by the communication phase and exploration phase respectively. Then we have

$$\mathbb{E}[R(T)] = \mathbb{E}[R_{comm}(T)] + \mathbb{E}[R_{expl}(T)] .$$

Recall the definition of $\mathcal{F}$, under event $\neg\mathcal{F}$ we can imply that $\text{UCB}_{i,k} \geq \mu_{i,k} \geq \text{LCB}_{i,k}$. And the regret caused by exploration can be bounded by

$$\mathbb{E}[R_{expl}(T)] \leq \mathbb{E}[R_{expl}(T) \mid \neg\mathcal{F}] + T\mathbb{P}(\mathcal{F}) .$$

For each epoch $s$, we define the set of player-arm pair $\mathcal{D}_s$ as the eliminated pair at epoch $s$.

For arm $(i,k) \in \mathcal{D}_s$, we have that it is not eliminated at epoch $s-1$, where each non-eliminated arm including $(i,k)$ has been selected at least $2^s$ number of times. Then conditioned on good event $\neg\mathcal{F}$, we have that

$$|\hat{\mu}_{i,k}(s) - \mu_{i,k}| \leq \sqrt{\frac{6\log T}{2^s}} .$$

Moreover, since the optimal player-arm pair (i', k') with max-min reward $\gamma^*$ is not eliminated, it is also selected at least $2^s$ number of times, we have that

$$|\hat{\mu}_{i',k'}(s) - \mu_{i',k'}| \leq \sqrt{\frac{6 \log T}{2^s}}.$$

Since sub-optimal pair $(i, k)$ is not eliminated, we have that $2\sqrt{\frac{6 \log T}{2^s}} \geq \mu_{i',k'} - \mu_{i,k} := \Delta_{i,k}$. Otherwise it must hold that $\text{UCB}_{i,k}(s) < \text{LCB}_{i',k'}(s)$ and (i,k) will be eliminated after $s - 1$ epoch.

Thus conditioned on $\neg \mathcal{F}$, we have that

$$2^s \leq 24 \log T / \Delta_{i,k}^2.$$

This implies that $\Delta_{i,k} \leq \sqrt{\frac{24 \log T}{2^s}}$, where $\Delta_{i,k} := \gamma^* - \mu_{i,k}$. Denote $n_{i,k}(T) = \sum_{t=1}^T \mathbb{1}\{(i, k) \in \arg\min_{(j,\ell) \in m_t} \mu_{j,\ell}\}$ as the number of times sub-optimal pair $(i, k)$ counts for the regret. Following the algorithm design, we know that the total rounds up to epoch $s$ is at most $(N^2 + K)2^{s+1}$, thus

$$\sum_{(i,k) \in \mathcal{D}_s} n_{i,k}(T) \leq (N^2 + K)2^{s+1}.$$

Denote $s_{\max}$ as the last epoch that eliminates sub-optimal pairs. We have that

$$s_{\max} \leq \log\left(24 \log T / \Delta^2\right).$$

Thus regret can be decomposed as

$$
\begin{aligned}
\mathbb{E}\left[R_{expl}(T) \mid \neg\mathcal{F}\right] &= \sum_{(i,k)} n_{i,k}(T)\Delta_{i,k} \\
&= \sum_{s=1}^{s_{\max}} \sum_{(i,k) \in \mathcal{D}_s} n_{i,k}(T)\Delta_{i,k} \\
&\leq \sum_{s=1}^{s_{\max}} (N^2 + K)2^{s+1}\sqrt{\frac{24 \log T}{2^s}} \\
&= \sum_{s=1}^{s_{\max}} 4(N^2 + K)\sqrt{6 \log T}\, 2^{\frac{s}{2}} \\
&\leq 4(N^2 + K)\sqrt{6 \log T}\, \frac{\sqrt{2}(1 - \sqrt{2}^{s_{\max}})}{1 - \sqrt{2}} \\
&\leq 4(N^2 + K)\sqrt{6 \log T}\, \frac{\sqrt{2}(1 - \sqrt{24 \log T / \Delta^2})}{1 - \sqrt{2}} \\
&\leq 164(N^2 + K) \log T / \Delta.
\end{aligned}
$$

The regret for the communication phase is bounded by the communication length times the number of phase. Since the length of phase grows exponentially we have that the total number of phase is less than $\log T$. Then the communication regret is

$$\mathbb{E}\left[R_{comm}(T)\right] \leq \gamma^* \frac{N(N-1)}{2\lfloor\frac{N}{2}\rfloor} \log T \leq \gamma^* N \log T.$$

Together with above results, we get the final regret bound. $\qquad \square$

## C. Proof for Section 4

We construct the special instance $\nu$ with $N \leq K$, the reward mean is designed as follows: For player $i \in [1, N-2]$, the reward mean of arms $k \in [i+2, N]$ is 0. For arm $k > N$, the reward mean is 0 for every player $i \in [N]$. For player $i \in [1, \lfloor \frac{N}{2} \rfloor]$, the reward mean of arms $k \in [1, i+1]$ is $\frac{1}{2} + 2\Delta$. For player $i \in [\lfloor \frac{N}{2} \rfloor, N - \lfloor \frac{N}{2} \rfloor + 1]$, the reward mean of arm $k = i$ is $\frac{1}{2} + \Delta$, the reward mean of arm $k = i+1$ is $\frac{1}{2} + 2\Delta$. For player $i \in [N - \lfloor \frac{N}{2} \rfloor + 2, N]$, the reward mean of arm $k \in [N - \lfloor \frac{N}{2} \rfloor + 3, \min\{i+1, N\}]$ is $\frac{1}{2} + 2\Delta$. Other player-arm pairs have reward means $\frac{1}{2}$. Denote $\nu_{i,k}$ as the distribution of rewards obtained when arm $k$ is matched to player $i$ in this environment. The special case is shown in Figures 2 and 3.

We denote the set of player-arm pairs $S_1 = \{(i,k) \mid \mu_{i,k} = 0\}$, $S_2 = \{(i,k) \mid i \in [N - \lfloor \frac{N}{2} \rfloor + 1, N], k \in [1, \lfloor \frac{N}{2} \rfloor]\}$.

We first show that the max-min value is $\frac{1}{2} + \Delta$. First, we can find a matching $m^* = \{(1,1), (2,2), \ldots, (N,N)\}$ with minimum reward $\frac{1}{2} + \Delta$. Second, if there exists a matching $m'$ with minimum reward $\frac{1}{2} + 2\Delta$, then by construction the player $i \in [\lfloor \frac{N}{2} \rfloor + 1, N - \lfloor \frac{N}{2} \rfloor + 1]$ must select arm $k = i+1$. To reach the reward $\frac{1}{2} + 2\Delta$ player $i \in [N - \lfloor \frac{N}{2} \rfloor + 2, N-1]$ must select arm $i + 1$, and then player $i = N$ fails to select any arm with reward $\frac{1}{2} + 2\Delta$. Thus the max-min value is $\frac{1}{2} + \Delta$.

### C.1. Proof for Theorem 4.1

*Proof.* For term $N_{i,k}(T)$, we apply the basic technique in lower bound proof. We consider the above instance $\nu$, universally consistent policy $\pi$, player-arm $(i,k) \in S_2$. Let us consider another instance $\nu'$ (which is specific to player $i$ and arm $k$) where $\nu'_{i',k'} = \nu_{i',k'}$ for all $(i', k') \neq (i, k)$, $\nu'_{i,k}$ such that $D(\nu_{i,k}, \nu'_{i,k}) \leq D_{\inf}(\nu_{i,k}, \frac{1}{2} + \Delta, \mathcal{P}) + \epsilon$ and $\mu'_{i,k} > \frac{1}{2} + \Delta$. Here $D_{\inf}(\nu, \mu, \mathcal{P}) = \inf_{\nu' \in \mathcal{P}}\{D(\nu, \nu') : \mu'_{i,k} > x\}$. Then the max-min value is $\mu'_{i,k}$ and the corresponding max-min player-arm is $(i, k)$.

For any event $A$ (and its complement $A^c$), applying Pinsker's inequality we have

$$D(\mathbb{P}_{\nu,\pi}, \mathbb{P}_{\nu',\pi}) \geq \log\left(\frac{1}{2(\mathbb{P}_{\nu,\pi}(A) + \mathbb{P}_{\nu',\pi}(A^c))}\right).$$

Now consider $A = \{N_{i,k}(T) \geq T/2\}$. Thus we have the regret:

1. In instance $\nu$ as $R(T, \nu, \pi) \geq \Delta \frac{T}{2} \mathbb{P}_{\nu,\pi}(A)$.

2. In instance $\nu'$ as $R(T, \nu', \pi) \geq (\mu'_{i,k} - (\frac{1}{2} + \Delta)) \frac{T}{2} \mathbb{P}_{\nu',\pi}(A^c)$.

As the only change in reward distribution happens in $(i,k)$ pair, from the divergence decomposition lemma (Lemma 18 in Sankararaman et al. (2021)), we have that

$$D(\mathbb{P}_{\nu,\pi}, \mathbb{P}_{\nu',\pi}) = D(\nu_{i,k}, \mu'_{i,k}) \mathbb{E}_{\nu,\pi}[N_{i,k}(T)] \leq \left(\epsilon + D_{\inf}(\nu_{i,k}, \frac{1}{2} + \Delta, \mathcal{P}) \mathbb{E}_{\nu,\pi}\right)[N_{i,k}(T)].$$

Then we have

$$\left(\epsilon + D_{\inf}(\nu_{i,k}, \frac{1}{2} + \Delta, \mathcal{P}) \mathbb{E}_{\nu,\pi}\right)[N_{i,k}(T)] \geq \log\left(\frac{1}{2(\mathbb{P}_{\nu,\pi}(A) + \mathbb{P}_{\nu',\pi}(A^c))}\right)$$
$$\geq \log\left(\frac{T \min(\mu'_{i,k} - (\frac{1}{2} + \Delta), \Delta)}{4(R(T, \nu, \pi) + R(T, \nu', \pi))}\right).$$

The final inequality holds as the policy $\pi$ is assumed to be universally consistent. Thus we have

$$\lim_{\epsilon \to 0} \lim_{T \to \infty} \in \frac{\mathbb{E}_{\nu,\pi}[N_{i,k}(T)]}{\log T} \geq \lim_{\epsilon \to 0} \frac{1}{\epsilon + D_{\inf}(\nu_{i,k}, \frac{1}{2} + \Delta, \mathcal{P})} = \frac{1}{D_{\inf}(\nu_{i,k}, \frac{1}{2} + \Delta, \mathcal{P})}.$$

For $\mathcal{P}$ be the class of Bernoulli rewards, we have $D_{\inf}(\nu_{i,k}, \frac{1}{2} + \Delta, \mathcal{P}) \leq \Delta^2/2$. Then we have $\mathbb{E}[N_{i,k}(T)] \geq 2 \log T / \Delta^2$. And thus the total regret is lower bounded by

$$R(T) \geq \left(\lfloor \frac{N}{2} \rfloor\right)^2 \frac{2 \log T}{\Delta}.$$

As for the $\Omega(K \log T/\Delta)$ lower bound, we can simply let $N = 1$ and this problem is reduced to the classic single-player multi-armed bandit problem, which has $\Omega(K \log T/\Delta)$ lower bound.

$\square$

## D. Proof for Technical Lemmas

**Lemma D.1.** *(Corollary 5.5 in Lattimore & Szepesvári (2020)) Assume that $X_1, X_2, \ldots, X_n$ are independent, $\sigma$-subgaussian random variables centered around $\mu$. Then for any $\varepsilon > 0$,*

$$\mathbb{P}\left(\frac{1}{n}\sum_{i=1}^{n} X_i \geq \mu + \varepsilon\right) \leq \exp\left(-\frac{n\varepsilon^2}{2\sigma^2}\right), \quad \mathbb{P}\left(\frac{1}{n}\sum_{i=1}^{n} X_i \leq \mu - \varepsilon\right) \leq \exp\left(-\frac{n\varepsilon^2}{2\sigma^2}\right).$$

### D.1. Proof for Lemma 4.2

We also divide $T$ rounds into two parts: if any pair in $S_1$ occurs in $m(t)$, or there is an empty matching pair, then $t \in \mathcal{T}_1$. If $m(t)$ contains no $S_1$ pairs but contains any $S_2$ pairs, then $t \in \mathcal{T}_2$. It is easy to verify that $\mathcal{T}_1$ and $\mathcal{T}_2$ do not intersect. Denote $T_1 = \mathbb{E}\left[|\mathcal{T}_1|\right]$ and $T_2 = \mathbb{E}\left[|\mathcal{T}_2|\right]$.

$$\begin{aligned}
R(T, \nu, \pi) &= \mathbb{E}\left[\sum_{t=1}^{T}\left(1 - \min_{i}\left\{(1 - C_i(t)) \cdot \mu_{i,k_i(t)}\right\}\right)\right] \\
&\geq (\frac{1}{2} + \Delta)\mathbb{E}\left[\sum_{t=1}^{T}\mathbb{1}\{t \in \mathcal{T}_1\}\right] + \Delta\mathbb{E}\left[\sum_{t=1}^{T}\mathbb{1}\{t \in \mathcal{T}_2\}\right] \\
&= (\frac{1}{2} + \Delta) \cdot T_1 + \Delta \cdot T_2.
\end{aligned} \tag{1}$$

Before analyzing term $T_2$, we first claim the following property of matching when no pair in $S_1$ is selected.

For term $T_2$, recall that it means the number of times $m(t)$ contains no $S_1$ pairs but any $S_2$ pairs. Then we can lower bound this term by

$$\begin{aligned}
T_2 &= \sum_{t=1}^{T}\mathbb{1}\{\forall(i', k') \in S_1, (i', k') \notin m(t) \text{ and } \exists(i, k) \in S_2, (i, k) \in m(t)\} \\
&\geq \sum_{t=1}^{T}\sum_{i=N-\lfloor\frac{N}{2}\rfloor+1}^{N}\sum_{k=1}^{\lfloor\frac{N}{2}\rfloor}\mathbb{1}\{\forall(i', k') \in S_1, (i', k') \notin m(t) \text{ and } (i, k) \in m(t)\} \\
&= \sum_{i=N-\lfloor\frac{N}{2}\rfloor+1}^{N}\sum_{k=1}^{\lfloor\frac{N}{2}\rfloor}\sum_{t=1}^{T}\mathbb{1}\{\forall(i', k') \in S_1, (i', k') \notin m(t) \text{ and } (i, k) \in m(t)\} \\
&= \sum_{i=N-\lfloor\frac{N}{2}\rfloor+1}^{N}\sum_{k=1}^{\lfloor\frac{N}{2}\rfloor}\sum_{t=1}^{T}\left(\mathbb{1}\{(i, k) \in m(t)\} - \mathbb{1}\{(i, k) \in m(t), \exists(i', k') \in S_1, (i', k') \in m(t)\}\right) \\
&= \sum_{i=N-\lfloor\frac{N}{2}\rfloor+1}^{N}\sum_{k=1}^{\lfloor\frac{N}{2}\rfloor}\sum_{t=1}^{T}\mathbb{1}\{(i, k) \in m(t)\} \\
&\quad - \sum_{i=N-\lfloor\frac{N}{2}\rfloor+1}^{N}\sum_{k=1}^{\lfloor\frac{N}{2}\rfloor}\sum_{t=1}^{T}\mathbb{1}\{(i, k) \in m(t), \exists(i', k') \in S_1, (i', k') \in m(t)\} \\
&\geq \sum_{i=N-\lfloor\frac{N}{2}\rfloor+1}^{N}\sum_{k=1}^{\lfloor\frac{N}{2}\rfloor}\sum_{t=1}^{T}\mathbb{1}\{(i, k) \in m(t)\} - \left(\lfloor\frac{N}{2}\rfloor\right)\sum_{t=1}^{T}\mathbb{1}\{\exists(i', k') \in S_1, (i', k') \in m(t)\}
\end{aligned}$$

$$= \sum_{i=N-\lfloor \frac{N}{2} \rfloor+1}^{N} \sum_{k=1}^{\lfloor \frac{N}{2} \rfloor} N_{i,k}(T) - \left( \lfloor \frac{N}{2} \rfloor \right) T_1 \,.$$

The first inequality is derived by Claim 1. The last inequality holds since at most $\lfloor \frac{N}{2} \rfloor$ player-arm pairs in $S_2$ can be simultaneously selected in one matching. And thus $T_1$ can be repeated count $\lfloor \frac{N}{2} \rfloor$ times, leading to

$$\sum_{i=N-\lfloor \frac{N}{2} \rfloor+1}^{N} \sum_{k=1}^{\lfloor \frac{N}{2} \rfloor} \sum_{t=1}^{T} \mathbb{1}\{(i,k) \in m(t), \exists (i',k') \in S_1, (i',k') \in m(t)\}$$

$$\leq \left( \lfloor \frac{N}{2} \rfloor \right) \sum_{t=1}^{T} \mathbb{1}\{\exists (i',k') \in S_1, (i',k') \in m(t)\} \,.$$

Therefore, we can lower bound the regret as

$$R(T,\nu,\pi) \geq (\frac{1}{2} + \Delta) \cdot T_1 + \Delta \cdot T_2$$

$$\geq (\frac{1}{2} - \lfloor \frac{N}{2} \rfloor \Delta) T_1 + \Delta \sum_{i=N-\lfloor \frac{N}{2} \rfloor+1}^{N} \sum_{k=1}^{\lfloor \frac{N}{2} \rfloor} \mathbb{E}\left[ N_{i,k}(T) \right] \tag{2}$$

$$\geq \Delta \sum_{i=N-\lfloor \frac{N}{2} \rfloor+1}^{N} \sum_{k=1}^{\lfloor \frac{N}{2} \rfloor} \mathbb{E}\left[ N_{i,k}(T) \right] \,,$$

where last inequality holds for $\Delta$ sufficiently small that $\Delta < 1/N$. This ends the proof.

### D.2. Proof for Lemma B.1

$$\mathbb{P}(\mathcal{F}) = \mathbb{P}\left( \exists 1 \leq t \leq T, i \in [N], k \in [K] : |\hat{\mu}_{i,k}(t) - \mu_{i,k}| > \sqrt{\frac{6 \log T}{N_{i,k}(t)}} \right)$$

$$\leq \sum_{t=1}^{T} \sum_{i \in [N]} \sum_{k \in [K]} \mathbb{P}\left( |\hat{\mu}_{i,k}(t) - \mu_{i,k}| > \sqrt{\frac{6 \log T}{N_{i,k}(t)}} \right)$$

$$\leq \sum_{t=1}^{T} \sum_{i \in [N]} \sum_{k \in [K]} \sum_{s=1}^{t} \mathbb{P}\left( N_{i,k}(t) = s, |\hat{\mu}_{i,k}(t) - \mu_{i,k}| > \sqrt{\frac{6 \log T}{s}} \right)$$

$$\leq \sum_{t=1}^{T} \sum_{i \in [N]} \sum_{k \in [K]} t \cdot 2 \exp(-3 \log T)$$

$$\leq 2NK/T \,,$$

where the second last inequality is due to Lemma D.1.

# E. Details of Experiments

We give the reward matrix of the experiments in Section 5 here. As for another reward matrix with shape $10 \times 15$, all reward means are uniformly sampled from $[0,1]$ with no other constraints. All experiments are conducted on CPU.

$$U_{4\times4} = \begin{bmatrix} 0.5 & 0.9 & 0.1 & 0.25 \\ 0.25 & 0.5 & 0.25 & 0.1 \\ 0.1 & 0.25 & 0.5 & 0.5 \\ 0.1 & 0.9 & 0.25 & 0.5 \end{bmatrix},$$

$$U_{10 \times 10} = \begin{bmatrix} 0.9 & 0.4 & 0.8 & 0.1 & 0.3 & 0.05 & 0.2 & 0.1 & 0.3 & 0.2 \\ 0.4 & 0.3 & 0.3 & 0.1 & 0.2 & 0.3 & 0.4 & 0.4 & 0.3 & 0.4 \\ 0.1 & 0.05 & 0.1 & 0.4 & 0.1 & 0.2 & 0.9 & 0.3 & 0.4 & 0.1 \\ 0.05 & 0.1 & 0.9 & 0.2 & 0.9 & 0.75 & 0.1 & 0.9 & 0.25 & 0.05 \\ 0.8 & 0.3 & 0.1 & 0.7 & 0.1 & 0.4 & 0.05 & 0.2 & 0.75 & 0.05 \\ 0.4 & 0.05 & 0.3 & 0.7 & 0.05 & 0.1 & 0.25 & 0.75 & 0.6 & 0.05 \\ 0.9 & 0.3 & 0.3 & 0.8 & 0.1 & 0.25 & 0.7 & 0.05 & 0.2 & 0.3 \\ 0.3 & 0.1 & 0.4 & 0.25 & 0.05 & 0.9 & 0.25 & 0.1 & 0.05 & 0.4 \\ 0.8 & 0.75 & 0.1 & 0.2 & 0.4 & 0.05 & 0.3 & 0.2 & 0.1 & 0.25 \\ 0.4 & 0.4 & 0.9 & 0.7 & 0.25 & 0.2 & 0.05 & 0.1 & 0.4 & 0.25 \end{bmatrix}.$$

## F. Details of Communication Phase

In this section, we provide the details of communication phase in Algorithm 1 if we send information by bit. As the communication order can be decided as soon as the number of players is given, we focus on the communication between any player $i \in \mathcal{N}$ and player $j \in \mathcal{N}$, where $i < j$ in epoch $s$ in Algorithm 3. If player $i$ and $j$ collide in arm $i$, they receive digit 1 from each other; if they collide in arm $j$, they receive digit 0 from each other, otherwise, their current digit is different from one another. By such process, player $i$ can receive $\hat{\mu}_{j,k}(s)$ and player $j$ can receive $\hat{\mu}_{i,k}(s)$ for all $k$.

---

**Algorithm 3** Communication phase in Decentralized Fair Elimination (for players $i$ and $j$)

---

1: Players $i$ and $j$ transfer $\hat{\mu}_{i,k}(s)$ and $\hat{\mu}_{j,k}(s)$ for every $k \in \mathcal{K}$ into binary data, $\tilde{\mu}_{i,k}(s)$ and $\tilde{\mu}_{j,k}(s)$, respectively.
2: **for** $k = 1, 2, \ldots, K$ **do**
3:    **for** $\tau = 1, 2, \ldots, L$ **do**
4:       **if** the $\tau$'s digit of $\tilde{\mu}_{i,k}(s)$ is 1 **then**
5:          Player $i$ selects arm $i$.
6:       **else**
7:          Player $i$ selects arm $j$.
8:       **end if**
9:       **if** the $\tau$'s digit of $\tilde{\mu}_{j,k}(s)$ is 1 **then**
10:      Player $j$ selects arm $i$.
11:      **else**
12:        Player $j$ selects arm $j$.
13:      **end if**
14:    **end for**
15: **end for**

---

