# OpenReview forum: "Optimal Algorithm for Max-Min Fair Bandit"
_ICML.cc/2025/Conference — ICML 2025 poster_

### Official Review · Reviewer_MKEL · 2025-02-27

**Overall Recommendation:** 4

**Summary:**

The authors study the max-min fair bandit optimization problem where the objective is to maximize the minimum reward achieved in a multi-player multi armed bandit instance. This paper designs a decentralized fair elimination algorithm that achieves an improved regret bound of $O((N^2+K)log(T)/\Delta)$. They provide a regret lower bound of $\Omega(\max(N^2, K) \log(T)/\Delta)$ which shows the tightness of the regret upper bound.  The algorithm relies on finding a lower bound of the max-min objective using the LCB indices of the arms, and uses this to eliminate arms with UCB lower than the aforementioned bounds. The non-eliminated arms are explored. When the optimal UCB based matching is not higher than the lower bound, the algorithm terminates with a resultant matching. Numerical simulations show the regret improvement achieved in this paper.

Edit after rebuttal: The authors provided clarifications to some of my questions. I maintain my positive score.

**Claims And Evidence:**

The claims seem well grounded by mathematical proofs to the best of my understanding.

**Essential References Not Discussed:**

Not that I know of.

**Ethical Review Concerns:**

Theoretical paper. So no data related ethical issues. The paper talks about fairness, but max-min fairness is well established concept and not controversial.

**Experimental Designs Or Analyses:**

Numerical experiments look valid, however it is somewhat anecdotal. But theoretical guarantees provide a better understanding of the merit of this paper.

**Methods And Evaluation Criteria:**

This is a theoretical paper. The methodology seems sound.

**Other Comments Or Suggestions:**

N/A

**Other Strengths And Weaknesses:**

Strength: The regret bounds improve the state-of-the-art. The regret bounds seem tight at least for the parameters $N,K, \Delta, T$.

Improvements: Maybe in future a fully instance dependent algorithm/analysis can be explored.

**Questions For Authors:**

- Do you assume knowledge of $\Delta$ in the collision based communication?

**Relation To Broader Scientific Literature:**

This improves the literature of max-min fairness in bandit learning.

**Theoretical Claims:**

I checked the upper bound claims at some detail. The claims seem correct at least order wise. The constants are not checked.

I checked the lower bound proof at a high level. This claims also makes sense. It is unlikely, but possible that missed some details around the lower bound proof.

---

> ### Author Rebuttal · Authors · 2025-03-31
>
> We thank the reviewer for your valuable and detailed comments. Please see our response below.
>
> Q1. Do you assume knowledge of $\Delta$ in the collision based communication?}
>
> In the collision-based communication discussed in Remark 1, we assume the algorithm knows the order of $\Delta$ such that the algorithm can use exactly $\log 1/\Delta$ rounds at each communication phase. However, when the target max-min matching is unique, we can design algorithm with increasing communication length and still achieves $\log T$ communication regret. Specifically, after $s$-th exploration phase, the algorithm enters the communication phase of length $O(s)$, which is determined by the estimation precision controlled of order $1/2^s$. Then the exploration phase will end if the precision order reaches $O(\Delta)$, and the corresponding communication length is $O(\log 1/\Delta)$, which matches the result of knowing $\Delta$.

---

> > ### Comment · Reviewer_MKEL · 2025-04-03
> >
> > I thank the authors for clarifying my doubts. A way to make the communication work for general case would be great, but I understand if that is out-of-scope for this work. Maybe the authors can add the above response in the main paper or appendix (as appropriate). I will maintain my positive score.

---

### Official Review · Reviewer_qQDX · 2025-03-10

**Overall Recommendation:** 4

**Summary:**

This paper studies the multi-player multi-armed bandit problem in a heterogeneous setting with collisions, focusing on max-min fairness. Instead of maximizing total rewards, the goal is to maximize the reward of the player who receives the lowest reward, ensuring fairness. The contributions are as follows: (i) propose a new algorithm to achieve optimal regret bound. (ii) provide a regret lower bound (iii) demonstrate their algorithm using synthetic datasets.


## update after rebuttal
After the rebuttal, I will maintain my score as the contribution of this work remains clear.

**Claims And Evidence:**

The claims are clear.

**Essential References Not Discussed:**

It would be beneficial to reference matching bandits. Although the objectives of matching bandits and MP-MAB differ, elimination-based algorithms have been explored in matching bandits, similar to the approach used in the proposed algorithm.

**Experimental Designs Or Analyses:**

Experimental designs are reasonable.

**Methods And Evaluation Criteria:**

The method and evaluation criteria make sense.

**Other Comments Or Suggestions:**

Typos:
In line 5 of algorithm 2:  mathcal{P}--> mathcal{K}_{m'}

**Other Strengths And Weaknesses:**

Strengths
1. The paper introduces a novel algorithm for achieving optimal regret.
2. The paper provides a lower bound for regret.
3. the paper demonstrates the algorithm using synthetic datasets.


Weaknesses
1. I cannot find a weakness.

**Questions For Authors:**

1. It is unclear how to explore the remaining $K - N$ arms when $ K - N < N $. How can $ m $ be constructed without causing collisions?

2. In the definitions of $\gamma^*$ and $ m^* $, does each matching instance for maximization exclude cases where collisions occur?

**Relation To Broader Scientific Literature:**

They propose a novel algorithm that can achieve optimal regret bound in MP-MAB, while the previous algorithm for it did not.

**Theoretical Claims:**

I reviewed the theoretical claims in the main paper but did not check the detailed proofs in the appendix. However, the arguments in the main part appear well-structured and logically sound.

---

> ### Author Rebuttal · Authors · 2025-03-31
>
> We thank the reviewer for your valuable and detailed comments. Please see our response below.
>
> Q1. It is unclear how to explore the remaining $K-N$ arms when $K-N< N$. How can be constructed without causing collisions?
>
> When $K-N < N$, we can also construct $K-N$ matchings for those remaining $K-N$ arms without collisions. Specifically, it can be considered as there are $N$ remaining arms but last $N - (K-N)$ arms are all eliminated. Then we can directly apply the Assign Exploration Algorithm (Algorithm 2). When some player $i$ will select arm with index exceeding the remaining $K-N$ arms, it will turn to select $m'_i$ (Line 11).
>
> Q2. In the definitions of $\gamma^\ast$ and $m^\ast$, does each matching instance for maximization exclude cases where collisions occur?
>
> Yes, for simplicity we exclude cases where collisions occur since the corresponding max-min reward is $0$ if there exists collisions. Moreover, the definition of matching $m$ is the set of edges without same player or arm, which avoids the case of collisions.

---

### Official Review · Reviewer_tZoy · 2025-03-10

**Overall Recommendation:** 4

**Summary:**

The authors consider the multi-player multi-armed bandit setting, where $N$ players each choose one of $K$ arms in a cooperative but decentralized manner. The authors propose a new algorithm for this setting with optimal regret, and the authors include the factors of $N,K$ and $\Delta$ in the regret bound. The authors also show a tight lower bound for this setting, showing that their algorithm is optimal. They also relax some assumptions from previous works, like bounded distributions, $N=K$, and requirements on the means of the arms.

**Claims And Evidence:**

Yes.

**Essential References Not Discussed:**

NA

**Experimental Designs Or Analyses:**

The experiments compare the new proposed method with the existing methods, and shows drastic improvement (which I assume comes from the improved dependency on N and K. I appreciate that the authors used the same mean reward matrix that was used in previous papers, which shows that they are not cherry picking situations where their algorithm performs well.

**Methods And Evaluation Criteria:**

The definition of regret the authors use is consistent with previous works (comparing to the best maxmin algorithm) and is the natural baseline for this problem.

**Other Comments Or Suggestions:**

While maxmin fairness has been studied in these previous two works, there are other common notions of fairness that have been studied for bandit settings like envy-freeness and proportionality [1] [2]. It might be interesting to mention these works and discuss how these notions of fairness differ from maxmin fairness in the cooperative distributed bandits setting. While proving anything new about these fairness notions is definitely beyond the scope of this submission, it could be also interesting to discuss if some of the algorithmic ideas from this paper could extend to fairness notions such as envy-freeness or proportionality.

[1] Yamada, Hakuei, et al. "Learning fair division from bandit feedback." International Conference on Artificial Intelligence and Statistics. PMLR, 2024.

[2] Procaccia, Ariel D., Ben Schiffer, and Shirley Zhang. "Honor among bandits: No-regret learning for online fair division." Advances in Neural Information Processing Systems 37 (2024): 13183-13227.

**Other Strengths And Weaknesses:**

Strengths
- The strength of this paper is that the authors give a new algorithm with a theoretical regret bound for this specific problem that improves on previous works by removing the $loglog(T)$ factor and drastically improving the dependency on N and K.
- The matching lower bound presented in this paper also provides a complete picture of the hardness of the setting.
- The theoretical tools used in the algorithm seem both new and interesting, especially the algorithmic ideas for exploration.
- The writing throughout is clear, and the intuition for the algorithm and proofs in the body are well-written and do a good job of communicating the main ideas.

Weaknesses
- One of the main weaknesses of the paper is that the setting studied is very specific (decentralized but cooperative bandits). One of the main selling points for the authors is that their algorithm performs significantly better in terms of N and K and therefore is more practical. However, the application discussed is not very convincing to me. In wireless networks, it seems likely that the different players are either non-cooperative or have full communication. I do understand that this work is primarily a theoretical contribution (and I strongly believe the paper does present some interesting new theoretical ideas). However, despite the two previous works on maxmin fair bandits, I am not sure how much broader impact this work will have on either the bandits or fairness literature.

**Questions For Authors:**

NA

**Relation To Broader Scientific Literature:**

The setting of maxmin fairness in cooperative but decentralized bandits has been studied before by multiple papers, and is a natural setting for fair multi-armed bandits. The results in this paper improve the previous best regret bounds by a factor of loglog(T), which itself is not a super interesting improvement. The improvement on the factors of N and K is more interesting, as that makes the algorithm significantly more practical and also (I am guessing) leads to the significantly better performance in the experimental setting. The technical ideas in the proposed algorithm seem new and interesting, though it is not immediately obvious to me that they are useful outside of this setting.

**Theoretical Claims:**

The theoretical claims in the body of the paper seem correct.

---

> ### Author Rebuttal · Authors · 2025-03-31
>
> We thank the reviewer for your valuable and detailed comments. Please see our response below.
>
> Q1. How much broader impact this work will have on either the bandits or fairness literature?
>
> We emphasize that the optimal exploration design in Algorithm 2 holds applicability beyond the current context. It can be effectively implemented for other Multi - Player Multi - Armed Bandit (MP - MAB) problems featuring heterogeneous rewards. In such scenarios, different player - arm pairs often require distinct exploration durations to achieve optimal regret. This exploration design isn’t limited to decentralized, cooperative bandit models. Instead, it can be extended to other setups, including those with communication channels or centralized control mechanisms.
>
> Moreover, our lower bound analysis remains valid across all multi-player bandit settings where collisions occur. Whether the system is decentralized or cooperative, our analysis provides a reliable foundation for understanding performance limits.
>
> Q2. Discuss if some of the algorithmic ideas from this paper could extend to fairness notions such as envy-freeness or proportionality.
>
> We are sincerely grateful to the reviewer for bringing attention to alternative fairness objectives explored in the bandit setting, such as envy - fairness and proportionality. In response, we offer a discussion on how the algorithmic concepts presented in this paper can be extended to encompass other fairness concepts.
> When fairness is examined from the players’ perspective—for instance, in the cases of max - min fairness and envy - fairness—these metrics can often be computed offline when the expected rewards are known. As a result, we can develop an elimination - based algorithm, similar to Algorithm 1. This approach initially distributes exploration efforts across different arms. Subsequently, once the learner has acquired sufficient confidence in its reward estimations, it computes the relevant fairness metrics.
> Moreover, our innovative exploration - allocation design is versatile and can be adapted to accommodate other fairness concepts, further expanding the applicability of our proposed algorithms in the realm of fair bandit algorithms.

---

> > ### Comment · Reviewer_tZoy · 2025-04-02
> >
> > Thank you for the detailed response! It could be interesting to include this discussion of the alternative fairness objectives in the final version of the paper.

---

### Official Review · Reviewer_R8eu · 2025-03-17

**Overall Recommendation:** 3

**Summary:**

This paper studies the learning problem of Multi-player multi-armed bandits. The reward model is heterogeneous. Also, if two distinct players choose the same arm, both players receive zero reward. The goal is to minimize max-min regret. This framework is interesting and useful for important real-world applications such as choosing channels in wireless systems. The paper is well-written.

## update after rebuttal: after the rebuttals, I keep my current score.

**Claims And Evidence:**

Yes.

**Essential References Not Discussed:**

No

**Experimental Designs Or Analyses:**

N/A

**Methods And Evaluation Criteria:**

Yes.

**Other Comments Or Suggestions:**

See previous box

**Other Strengths And Weaknesses:**

Strength: The construction of problem-instance for proving regret lower bound is interesting.

Weakness: I highly recommend to adding instance-independent regret bounds as $\Delta$ could be quite small. Also, to evaluate the tightness of the proposed algorithm, it is also nice to have minimax regret lower bounds. Last, elimination-based algorithm has lots of downsides, developing UCB or Thompson Sampling-based algorithms would be more useful.

**Questions For Authors:**

Questions: In Section 3.1, why $LCB_{j,k}(s)$ and $UCB_{j,k}(s)$ use the statistics of $(i,k)$?

**Relation To Broader Scientific Literature:**

The theoretical framework in this work is useful for many important real-world applications

**Theoretical Claims:**

Yes. I took a  look at the proofs in the appendix.

---

> ### Author Rebuttal · Authors · 2025-03-31
>
> We thank the reviewer for your valuable and detailed comments. Please see our response below.
>
> Q1. It is nice to have minimax regret upper/lower bound.
>
> We are grateful to the reviewer for highlighting the significance of deriving the minimax regret bound. This bound is crucial as it showcases the algorithm’s performance in scenarios where the minimum gap $\Delta$ is extremely small. In our work, we derived the instance-dependent regret upper and lower bounds, which aligns with common practices in the multi-player multi-armed bandit literature. Deriving the minimax regret upper and lower bounds represents an interesting and valuable direction for future research. We look forward to exploring this area further. The discussion of alternative approach to derive the minimax regret bound will be added in the updated version.
>
> Q2. Developing UCB or TS-based algorithms would be more useful.
>
> We recognize that algorithms based on UCB and TS are more adaptive compared to the elimination-based algorithm. The reason lies in the fact that once an arm is eliminated in the elimination - based algorithm, it will no longer be explored. However, it is important to note that in the heterogeneous multi-player bandit setting, the elimination-based algorithm excels at distributing exploration efforts among different players in a round-robin manner. This is a crucial step in conflict resolution and adaptation to a decentralized environment. In contrast, UCB or TS-based algorithms are more prone to encounters of collisions and non-uniform exploration patterns among players. Furthermore, designing a decentralized UCB or TS-based algorithm poses significant challenges. The absence of a platform to allocate arms in each round makes it difficult to implement such algorithms in a decentralized context.
>
> Q3. In Section 3.1, why $\text{UCB}\_{i, k}(s)$ and $\text{LCB}\_{i,k}(s)$ use the statistics of $(i,k)$?
>
> Since this paper studies the heterogeneous multi-player multi-armed bandit setting, each player-arm pair shares an expected reward $\mu\_{i,k}$. Thus we need to design $ \text{UCB}\_{i,k}(s)$ and $\text{LCB}\_{i,k}(s)$ in terms of $(i,k)$ to control the confidence radius of estimation $\hat{\mu}\_{i,k}$.

---

### Decision · Program_Chairs · 2025-05-01

**Decision:**

Accept (poster)

**Comment:**

This paper studies max-min fair-bandit optimization: the objective is to maximize the minimum reward achieved in a multi-player multi-armed-bandit instance. The rewards are fully heterogeneous; when two (or more) players compete for the same arm in a round, they “collide” and get zero reward; this models, e.g., channel-access in wireless networks. The paper develops a decentralized fair elimination algorithm that achieves a significantly-improved regret upper bound of O((N^2 + K) (log T) / Delta), where there are N players competing for K arms in T rounds, and Delta in (0,1) is the minimum reward gap: a matching regret lower bound is also shown. (The earlier dependence on 1/Delta was exponential.) The algorithm relies on finding a lower bound of the max-min objective using the LCB (lower confidence bound) indices of the arms and uses this to eliminate arms with “low” UCB (upper confidence bound); the non-eliminated arms are explored. Numerical simulations show the regret improvement promised.

Both the optimal algorithm and the reasonable formulation make this a paper suitable for the conference.